# Understanding the Role of Autophagy in Cancer Formation and Progression Is a Real Opportunity to Treat and Cure Human Cancers

**DOI:** 10.3390/cancers13225622

**Published:** 2021-11-10

**Authors:** Simone Patergnani, Sonia Missiroli, Giampaolo Morciano, Mariasole Perrone, Cristina M. Mantovani, Gabriele Anania, Francesco Fiorica, Paolo Pinton, Carlotta Giorgi

**Affiliations:** 1Laboratory for Technologies of Advanced Therapies, Department of Medical Sciences, University of Ferrara, 44121 Ferrara, Italy; simone.patergnani@unife.it (S.P.); sonia.missiroli@unife.it (S.M.); giampaolo.morciano@unife.it (G.M.); mariasole.perrone@unife.it (M.P.); mcristinamantovani@gmail.com (C.M.M.); 2Department of Medical Sciences, Section of General and Thoracic Surgery, University of Ferrara, 44121 Ferrara, Italy; gabriele.anania@unife.it; 3Department of Radiation Oncology and Nuclear Medicine, AULSS 9 Scaligera, Ospedale Mater Salutis di Legnago, 37045 Verona, Italy; francesco.fiorica@aulss9.veneto.it

**Keywords:** cancer, autophagy, therapy, tumor suppression, clinical trials

## Abstract

**Simple Summary:**

The modulation of autophagy represents a potential therapeutic strategy for cancer. More than one hundred clinical trials have been conducted or are ongoing to explore the efficacy of autophagy modulators to reduce the tumor growth and potentiate the anti-cancer effects of conventional therapy. Despite this, the effective role of autophagy during tumor initiation, growth, and metastasis remains not well understood. Depending on the cancer type and stage of cancer, autophagy may have tumor suppressor properties as well as help cancer cells to proliferate and evade cancer therapy. The current review aims to summarize the current knowledge about the autophagy implications in cancer and report the therapeutic opportunities based on the modulation of the autophagy process.

**Abstract:**

The malignant transformation of a cell produces the accumulation of several cellular adaptions. These changes determine variations in biological processes that are necessary for a cancerous cell to survive during stressful conditions. Autophagy is the main nutrient recycling and metabolic adaptor mechanism in eukaryotic cells, represents a continuous source of energy and biomolecules, and is fundamental to preserve the correct cellular homeostasis during unfavorable conditions. In recent decades, several findings demonstrate a close relationship between autophagy, malignant transformation, and cancer progression. The evidence suggests that autophagy in the cancer context has a bipolar role (it may act as a tumor suppressor and as a mechanism of cell survival for established tumors) and demonstrates that the targeting of autophagy may represent novel therapeutic opportunities. Accordingly, the modulation of autophagy has important clinical benefits in patients affected by diverse cancer types. Currently, about 30 clinical trials are actively investigating the efficacy of autophagy modulators to enhance the efficacy of cytotoxic chemotherapy treatments. A deeper understanding of the molecular pathways regulating autophagy in the cancer context will provide new ways to target autophagy for improving the therapeutic benefits. Herein, we describe how autophagy participates during malignant transformation and cancer progression, and we report the ultimate efforts to translate this knowledge into specific therapeutic approaches to treat and cure human cancers.

## 1. Introduction

Autophagy is a main catabolic mechanism of the cell and refers to an evolutionary process by which cellular components and damaged organelles are degraded or recycled through lysosomal activity. Autophagy contributes to preserve cellular homeostasis and provides the cells the ability to adapt to stressful conditions and prevent cellular damage and cell death. Autophagy not only occurs as a response to various stress signals but also happens at basal levels where it regulates the cellular and tissue development and cell survival/death events [1]. Thanks to these features, today, autophagy is a tightly regulated mechanism involved in diverse pathologies, in particular in cancer, where more than 100 clinical trials associated with autophagy modulation have been conducted or are active. Despite the majority of these clinical trials expecting to induce a blocking of the autophagy machinery, the effective role of autophagy in cancer still remains controversial. For some malignancies, autophagy is a cytoprotective mechanism that increases the tumor cell viability and growth. Cancer cells within tumors are characterized by hypoxic conditions and nutrient deprivation. Thanks to the recycling of cytoplasmic elements provided by autophagy, cancer cells can survive these stressful conditions. Furthermore, autophagy has also been found to reduce the cellular stress induced by chemotherapy agents. On the other hand, autophagy is also described as a mechanism for the suppression of cancer. In this case, autophagy eliminates dangerous cellular elements, preserves the genomics stability, and activates the immunoresponse processes toward malignant transformation. Overall, a deep understanding of the connections that exist between autophagy and cancer, together with a clear identification of the molecular mechanisms characterizing this axis, are crucial to develop an appropriate therapeutic approach based on autophagy modulation in cancer. This review aims to give an intelligible overview of the role of autophagy in cancer formation and progression, describe the complex dual role of autophagy in tumor survival or cell death, summarize the current therapeutic strategies for the treatment of tumors, and, finally, report the most promising clinical trials with autophagy modulators.

## 2. General Aspects of Autophagy and Molecular Mechanisms 

The term autophagy (Greek, “self-eating”) refers to a tightly evolutionary self-degradation system that is essential for the maintenance of the physiological homeostasis that occurs in response to several cellular and environmental stresses [2]. Over the past ten years, substantial research has been carried out in regard to understanding the regulation and molecular mechanisms of autophagy, and considerable interest has emerged in this area of research, enough to confer the Nobel Prize in Medicine or Physiology to Yoshinori Oshumi for his groundbreaking work on the autophagic mechanism [3].

In mammalian cells, at least three distinct forms of autophagy—macroautophagy, microautophagy, and chaperone-mediated autophagy (CMA)—can be distinguished based on the mode of cargo delivery to lysosomes. During microautophagy, the lysosomal membrane can directly invaginate to trap cytosolic cargo for degradation; this process can include intact organelles and can be defined in different terms, such as micromitophagy (for mitochondria), microlipophagy (for lipid droplets), and micropexophagy (for peroxisomes) [4]. CMA uses chaperones to identify cargo proteins containing a specific pentapeptide-targeting motif (KFERQ), which is recognized by heat shock cognate 71 kDa protein (HSC70) in the cytosol, that are targeted and bound to the lysosomal membrane protein LAMP2A [5]. In contrast, macroautophagy (hereafter referred to as autophagy) involves the sequestration of cytoplasmic cargo by de novo double-membrane vesicles (named autophagosomes) that ultimately fuse to lysosomes for degradation [6].

Autophagy is a highly catabolic process, conserved from yeast to humans, used by eukaryotic cells to maintain cellular homeostasis and cellular and organellar quality control in response to multiple forms of stress, including energy or nutrients deprivation, hypoxia, oxidative stress, endoplasmic reticulum (ER) stress, and infection [7,8,9]. 

From a molecular point of view (Figure 1), autophagy can be divided into four critical steps: initiation, nucleation, maturation, and degradation, strictly orchestrated by at least 37-autophagy-related proteins (ATGs), that deliver cytoplasmic cargo to the lysosome for degradation. Two nutrient-responsive kinases, 5′ AMP-activated protein kinase (AMPK) and mechanistic target of rapamycin (MTOR), are the main regulators (positive and negative, respectively) of autophagy and quickly respond to external stimuli, nutrient fluctuations, and phosphorylation/dephosphorylation events. AMPK is a heterotrimeric protein that acts as a monitor of intracellular energy levels by controlling the AMP/ATP ratio. AMPK also controls the cellular processes that increase the ATP amounts (such as glycolysis and β-oxidation), controls the fatty acids and cholesterol synthesis, and regulates the ATP-consuming mechanisms [10,11]. MTOR is a serine/threonine protein kinase belonging to the phosphatidylinositol kinase-related kinase (PIKK) family. Inside the cell, MTOR is found in two distinct complexes, MTORC1 and MTORC2, but only MTORC1 regulates the cellular metabolism and responds to external and internal signals, such as amino acids, glucose, and growth factors [12]. Apart from MTOR itself, MTORC1 is composed of various regulatory subunits (including Raptor, MLST8, PRAS40, and DEPTOR) [13], and, under physiological conditions, MTORC1 results in activity and represses the autophagy machinery by phosphorylating the unc-51-like kinase 1 (ULK1) serine threonine kinase complex (consisting of ULK1, ATG13, RIB-inducible coiled-coil protein 1 (FIP200), and ATG101) at Ser-757, Ser-638, and Ser-758 residues [14,15]. The inactivation of MTORC1 causes the dephosphorylation of the target ribosomal protein S6 kinase (p70S6K) and the translation initiation factor 4E binding protein-1 (4E-BP1), provoking a downregulation of the cellular protein synthesis translation and the activation of the autophagy-dependent self-digestive process. In this latter event, the AMPK inhibits MTORC1 directly by phosphorylating Raptor [16] and indirectly via the activation of the tuberous sclerosis 2 (TSC2) complex [17] to determine the dephosphorylation of ULK1 at Ser-757, Ser-638, and Ser-758 residues. Then, the AMPK phosphorylates ULK1 and ATG13 in specific amino acid residues (Ser-555, Ser-777, Ser-317, Ser-467) to activate the ULK1 complex [14].

The ULK1 complex activation represents the initiation step of autophagy [18] and triggers the nucleation of the phagophore by phosphorylating and activating class III phosphatidylinositol 3 kinase (PI3KC3) complexes. The core of the PI3KC3 complex consists of class III PI3K, BECN1 [19], vacuolar protein sorting 34 (VSP34) [20], and general vesicular transport factor (p115) [21], then it can be associated with both ATG14 [22] and the activating molecule in BECN1-regulated autophagy protein 1 (AMBRA1) (PI3KC3-C1) [23] or UV radiation resistance-associated gene protein (UVRAG) (PI3KC3-C2) [19], which both, in turn, generate local phosphatidylinositol-3-phosphate (PIP3) production at a characteristic ER structure called the omegasomes, the membranous regions that elongate to generate the phagophores [24]. Both the initiation and nucleation steps promote the formation of the autophagic vesicle membrane that can also include membranes from ER-mitochondria and ER-plasma membrane contact sites, mitochondria, recycling endosomes, and Golgi complex [25,26], as well as ATG9-containing vesicles. PIP3 then recruits, through the PIP3-binding domain, WD repeat domain phosphoinositide-interacting proteins (WIPI) and zinc-finger FYVE domain-containing protein 1 (DFCP1) to promote membrane elongation [27]. This process represents the transition from omegasomes to the phagophores.

Two ubiquitin (Ub)-like conjugation systems are necessary for autophagosome formation (maturation step). One system involves the covalent conjugation of the Ub-like mammalian ATG12 to ATG5, which, together with RAB37 [28], further establishes a complex with ATG16, which, in turn, links to WIPI2 [27] and associates with the phagophore membrane. Although ATG7 and ATG5 represent fundamental proteins for autophagy execution, recent investigations have unveiled that autophagy may also occur in *Atg5* or *Atg7*-deficient cells. In this case, autophagy is termed “alternative autophagy” and the elongation process depends on the activity of RAB9 GTPase [29]. The second pathway, the ATG8 family proteins, includes microtubule-associated protein 1A/B light-chain 3A (LC3) and the GABARAP (gamma-aminobutyric acid receptor-associated protein) subfamily to membrane-resident phosphatidylethanolamine (PE) thanks to the sequential action of the proteases ATG4 and ATG3 [30]. These complexes, both regulated by the E1-like enzyme ATG7, bring the conversion of the soluble form of LC3 (LC3-I) to the lipidated autophagic vesicles-associated form, known as LC3-II. LC3-II is one of the most commonly used markers of autophagy as it migrates faster than LC3-I by electrophoresis detection; additionally, green fluorescent protein-LC3 fusion protein can be used to detect autophagosomes shifting from diffuse to punctuate staining by microscopy [2]. The lipidation process of LC3 is critical for the expansion and closure of the phagophore. However, during this process, it has been unveiled that the association of WIPI4 with ATG2, which together recruit ATG9, is essential. Indeed, the WIPI4–ATG2 complex moves to the ER and mediates the tethering between the ER and phagophore to regulate the lipid transfers necessary for the expansion of the phagophore [31]. Furthermore, ATG9, whose recruitment to ER is determined by the WIPI4–ATG2 complex, is equally important for the lipid and membrane supply to the phagophore [32,33].

The closure of the autophagic vesicle engages members of the endosomal sorting complex required for transport (ESCRT), in particular the charged multivesicular body protein 2A (CHMP2A) and the vacuolar protein sorting-associated-4 (VPS4) [34]. Both proteins move on the outer leaf of the autophagic vesicle, but, meanwhile, CHMP2A is responsible for the closure of the autophagosome, and VSP4 determines the disassembly of ESCRT molecules. Consistent with this, the genetic inhibition of CHMP2A and VPS4 alters the closure of the vesicle and delays fusion with the lysosome [34].

The newly formed double-membrane vesicle, just termed autophagosome, traps the engulfed cytosolic material, recruited by cargo receptors, such as SQSTM1/P62 and NBR1, as autophagic cargo destinated for degradation [35]. Then, the autophagosome is transported along microtubules to the perinuclear region to fuse with lysosomal membranes and form an autophagolysosome. This event requires the changing in lysosomal pH, tethering factors, such as the multiprotein homotypic fusion and vacuole protein sorting (HOPS) complex, RAB7, and small GTPase, along with SNAREs proteins (syntaxin 17 (STX17) and synaptosomal-associated protein 29 (SNAP29) in the autophagosome and vesicle associated membrane protein 8 on the lysosome (VAMP8) [36,37]. Emerging evidence suggests that other proteins, such as the adaptor protein EPG5 and members of the LC3/GABARAP family, are involved in the autophagosome–lysosome fusion event [38,39]. 

The last phase encompasses the degradation of the autophagic cargo by acidic hydrolases in the lysosome, and the resulting breakdown products are released back to the cytoplasm to provide new sources of energy in response to the nutritional needs of the cell (Figure 1). 

Autophagy also exists in multiple variant forms that are characterized to sequester and degrade specific intracellular elements, such as proteins (proteinphagy), lipid droplet (lipophagy), xenobiotics (xenophagy), and organelles. Probably, the most studied selective form of autophagy toward an intracellular organelle is mitophagy, the autophagic removal of autophagy, which is an essential mitochondrial quality control mechanism [40]. Mitophagy may occur under physiological conditions (development, aging, and differentiation), it may be induced by diverse harmful conditions, such as loss of mitochondrial functioning, excessive reactive oxygen species (ROS) production, hypoxia, and accumulation of unfolded proteins, and it is involved during the pathogenesis of a number of human diseases, including neurodegeneration [41], cancer [42], and cardiovascular disease [43]. The possibility that, inside a cell, mitochondria could be removed was suggested in late 1915 [44] and confirmed in 1962 when, by using electron microscopy, mitochondrial fragments were identified in liver lysosomes [45]. However, the molecular mechanisms at the basis of mitophagy started to be unveiled in the early 2000s when it was demonstrated that, during the differentiative process, red blood cells [46,47,48] eliminate their mitochondria by a mechanism orchestrated by the OMM resident NIP3-like protein X (NIX/BNIP3L), which, thanks to a WXXL-like motif facing the cytosol, can bind LC3 and GABARAP proteins to recruit isolation membranes to mitochondria [49,50]. FUN14 domain-containing protein 1 (FUNDC1) represents another OMM protein that acts as a mitophagy receptor in response to determinate stimuli [51]. Under normal physiological conditions, SRC kinase phosphorylates FUNDC1. Upon hypoxia, SRC becomes inactivated and FUNDC1 is subsequently dephosphorylated. This results in an increase in association with LC3 and the incorporation of the damaged mitochondria into the autophagic vesicle [51]. However, probably, the best characterized molecular axis regulating mitophagy in mammals is the PTEN-induced kinase 1 (PINK1) and Parkinson protein 2 (parkin). In functional mitochondria, PINK1 is imported into the mitochondria in the IMM, processed by several protease, and then degraded by the proteasome. Following mitochondria uncoupling, PINK1 accumulates and stabilizes on the OMM where it phosphorylates Parkin to change its conformation, recruits Parkin on the mitochondria, and activates Parkin into an active phospho-Ub-dependent enzyme [52,53,54,55]. In this state, Parkin determines the ubiquitinization of diverse OMM proteins and the consequent recruitment of the Ub-binding autophagic receptors LC3, NBR1, MDP52, TAX1BP1 (TBK1), and p62/Sequestome [56,57]. Interestingly, it has been proven that, during Parkin-mediated mitophagy, the OMM results ruptured [55,58,59], and the IMM is potentially exposed to interaction with the autophagic vesicle. A recent work identifies the IMM protein prohibitin 2 (PHB2) as a crucial mitophagy receptor and demonstrates that PHB2 binds the LC3-interaction region (LIR) domain following the mitochondrial depolarization and proteasome-dependent rupture of the OMM [60]. Another IMM protein responsible for the regulation of mitophagy is cardiolipin (CL). Following mitophagy stimuli, CL translocates to the OMM and acts as a signal for the identification and elimination of harmed mitochondria [61]. Furthermore, it has been demonstrated that this process is facilitated by the fact that an LC3 protein possesses a CL-binding site [61]. Recent investigations also account for mitochondrial matrix proteins having an important role in mitophagy regulation. An example may be found in the nod-like receptor (NLR) family member NLRX1. NLRX1 contains a LIR domain, and, upon infection with the pathogen *Listeria*, oligomerizes to induce the binding of the LIR motif to LC3 to activate mitophagy [62]. Recently, two other mitochondrial matrix resident proteins have been associated with mitophagy regulation and activation in yeast. Consistently, a knockout of both mitochondrial kinases Pkp1 and Pkp2 abolishes the mitophagy program [63].

## 3. Autophagy in Health and Disease

The discovery of autophagy has emerged as a breakthrough in both physiological and pathological conditions. Numerous studies and different genetic approaches have clearly pointed out the implications of autophagy in several diseases, including cancer, neurodegeneration, cardiovascular and pulmonary diseases, lysosomal disorders, diabetes, and obesity. Furthermore, autophagy is widely implicated in physiological responses to exercise, microbial pathogenesis, and aging [64,65]. Autophagy is defined as an evolutionarily conserved process that assists the cells to adapt to a myriad of stress conditions, providing a pool of amino acids by demolishing proteins and peptides. Autophagy function, at the basal level, is to regulate the intracellular conditions through the cytoplasmic turnover of proteins and organelles. Hence, autophagy, as a cytoprotective mechanism, maintains cellular homeostasis, thereby enabling the cells to stride past crisis situations. Nonetheless, there are specific situations in which excessive or imbalanced autophagy may be deleterious for the cell, be associated with cellular toxicity, and potentially to the development and progression of pathological conditions. In the following sections, we briefly consider the role of autophagy in the most common human diseases.

### 3.1. Autophagy and Neurodegeneration

The neurodegenerative diseases are age-dependent, hereditary, or sporadic disorders that are manifested by the progressive loss of neural functions. The common bases in their pathogenesis are mitochondrial dysfunction and the accumulation of protein aggregates that can be avoided through autophagy degradation [66]. Hence, the activation of autophagy could play a crucial role in neurodegenerative therapeutics, providing a substantial platform in the field of medical science [67]. One of the important features of Parkinson’s disease (PD) is the presence in the nucleus of the neurons of Lewy bodies, which consist of an insoluble aggregated protein called α-synuclein. This protein is susceptible to degradation through chaperone-mediated autophagy (CMA) [68]. Nevertheless, in familial PD, the lysosomes are unable to take up mutant α-synuclein due to its very high affinity with the LAMP2A, a lysosomal receptor. The high-binding affinity does not allow the substrate to be translocated properly and blocks the uptake of substrates for CMA, thus preventing degradation [69]. PD is also characterized by a mutation in parkin and PINK1 genes. These two proteins regulate the mobilization of dysfunctional mitochondria to the autophagosome for turnover (i.e., mitophagy) [70,71,72]. Furthermore, mutations in the Parkin gene alone are also responsible for the establishment of PD; this mutation does not allow the protein to ubiquitinate voltage-dependent anion channel 1 (VDAC1) [56], resulting in the non-clearance of damaged mitochondria [70]. Alzheimer’s disease (AD) is characterized by the accumulation of two proteins: Tau and amyloid β-peptide (Aβ). Autophagy is required to remove these proteins and thus suppress neurodegeneration [73,74,75]. Moreover, the autophagosome containing γ-secretase and related enzymes involved in the generation of Aβ from precursor forms may provide a source of Aβ under conditions of impaired autophagosome–lysosome fusion [76]. Interestingly, downregulated levels of the markers of autophagy processes (including mitophagy) have been detected in several AD human samples, such as brains, serum, and cerebrospinal fluids [77,78,79], suggesting that autophagy is an initial adaptive response in neurodegeneration due to the pathological accumulation of substrates, which may represent a failed repair mechanism that also contributes to disease progression. Consistent with this, the reactivation of the autophagy process reduces cognitive impairments and Aβ accumulation and improves synaptic function and learning memory in AD animal models [80]. Recently, it has been shown that autophagy plays a fundamental role in multiple sclerosis (MS) too. Autophagy and mitophagy markers’ levels are significantly increased in the biofluids of MS patients during the active phase of the disease, indicating the activation of these processes. Moreover, autophagy inhibitors improved myelin production and normalized axonal myelination [41]. 

### 3.2. Autophagy and Microbial Adaptations

During infection, autophagy contributes to the immune response by degrading intracellular bacteria and viruses (i.e., xenophagy) [81,82,83,84,85]. On the other hand, autophagy takes part in the suppression of inflammation, including the downregulation of both interferon responses to viral infection and proinflammatory cytokine production to invading pathogens. Furthermore, it inhibits the maturation and secretion of proinflammatory cytokines dependent on inflammasomes, such as interleukin (IL)-1β and IL-18, through the preservation of mitochondrial function [86]. The autophagic process may also regulate an unconventional pathway for the secretion of cytokines (e.g., IL-1β) [87]. Autophagy intervenes not only in antiviral and antimicrobial responses but can also play crucial roles in adaptive immune responses, such as antigen presentation and lymphocyte development [88]. Nedjic et al. showed the high constitutive expression of autophagic proteins in thymic epithelial cells [89]. Their study indicates that autophagy focuses the MHC-II-peptide repertoire of thymic epithelial cells on their intracellular milieu, contributing to T-cell selection and the generation of a self-tolerant T-cell repertoire.

### 3.3. Autophagy in Cardiovascular Diseases

Alterations in autophagy have been associated with heart diseases, including cardiomyopathies, cardiac hypertrophy, ischemic heart disease, heart failure, and ischemia–reperfusion injury [90]. For example, genetic deficiency in LAMP2 causes cardiomyopathy known as Danon’s disease [91]. In these patients, cardiomyocytes with evidence of mitochondrial dysfunction have an increased number of autophagosomes in the same way as cardiac tissue from patients with heart failure [92]. Mice with the specific deletion of ATG5 display cardiac hypertrophy and contractile dysfunction [93,94]. A similar cardiac hypertrophy has been unveiled in mice overexpressing miRNA (miR)-199, a miR that blocks autophagy by activating MTORC1 [95]. Autophagy has also shown to have beneficial effects in the context of cardiovascular diseases. Becn1^+/−^ mice, in which autophagy is attenuated, have a greater resistance to cardiac damage following reperfusion than WT mice [96], and, in a model of pressure overload, it reduces the pathological cardiac remodeling [97]. The administration of miR-188 that induces ATG7 downregulation limits infarct size in a murine model of myocardial infarction [98]. However, probably, it is mitophagy to cover a main role in cardiovascular disease. Mitochondria constitute about 30–40% of the cardiomyocyte volume, and the cardiomyocyte function is related to the mitochondrial status. Preserving a healthy mitochondrial population is of fundamental importance for cardiac tissue. As a demonstration of this, Parkin^−/−^ mice exhibited increased cardiac damage and have a reduced survival rate due to a diminished mitophagy activity. Furthermore, mice bearing Parkin deletion display perinatal cardiomyopathy and premature death [99]. In line with this, the mitophagy receptor FUNDC1 has been found to protect the heart from ischemia-reperfusion (I/R) injury and improves the mitochondrial functioning. In addition, a recent work demonstrates that, during calcific aortic valve stenosis (CAVS), although the autophagy and mitophagy pathways are increased in CAVS, their further increase helps to counteract the dangerous phenotype. Indeed, treatment with rapamycin increases these catabolic pathways and reduces the cell death and excessive calcification of aortic valves [43]. However, the also detrimental effect of mitophagy has been observed in a cardiovascular context. Indeed, mice lacking the mitophagy regulator BNIP3 have reduced myocardial damage and preserve cardiac functions during I/R [100]; meanwhile, the overexpression of BNIP3 increases infarct size and induces cardiomyocyte apoptosis [100].

### 3.4. Autophagy in Diabetes and Tissue Metabolism

In diabetes, increased adiposity and insulin resistance are primarily attributed to defective mitochondria characterized by impaired beta-oxidation, accumulation of lipids, oxidative stress, and, hence, mitochondrial damage. Autophagy, in particular mitophagy, at this stage, eliminates oxidative stress and damaged mitochondria, thus protecting the development of insulin resistance and growth in adiposity [92]. The unfolded protein response (UPR) of pancreatic beta cells is regulated by autophagy. It has been observed that the autophagy-deficient pancreatic cells are susceptible to ER stress, which, in turn, is involved in the progression of diabetes. Hence, it has been found that the autophagy-deficient mice bred with obese mice develop severe diabetes with the reduction in beta cell survival and accumulation of ROS [101]. Autophagy regenerates and releases amino acids, lipids, and other metabolic precursors, which may have different effects on tissue metabolism. Interestingly, it has been demonstrated that exercise increases the autophagy turnover and mitochondrial fission in type 2 diabetes, with an increase in the levels of ATG7 and p62/SQSTM1 and decrease in LC3-II protein [102]. Furthermore, mutations of p62/SQSTM1 have been linked to Paget’s disease too, a disorder of bone metabolism [103].

### 3.5. Role of Autophagy in Aging

With the onset of aging, autophagy gradually diminishes, consequently leading to the reduced formation of autophagy vacuoles and inopportune fusion between vacuoles and lysosomes. This finally provokes an accumulation of autophagy vacuoles in old tissues due to a significant impairment in the protein flux [104]. The most studied pathway that is implicated in longevity is the IGF-1 pathway [105]. The IGF-1 pathway includes PtdIns 3-kinase, tyrosine kinase receptor, and Akt/PKB. Interestingly, the deregulation of the Akt/PKB pathway may induce autophagy, confirming the linking of the IGF-1 pathway to autophagy. Supporting the role of autophagy in aging, a report demonstrated that the lifespan could be extended when TOR kinase in *C. elegans* is depleted [106]. Furthermore, it has been shown that some autophagic genes (i.e., ATG5, ATG7, and BECN1) are downregulated in the brains of old persons as compared with young persons and, conversely, upregulated in persons with AD [107]. Autophagy exerts a multifaced influence on the initiation and progression of cancer, as well as on the efficacy of therapeutic interventions in this disease, but this topic will be discussed in detail in the following sections. Taken together, these data revealed that the impact of the autophagic molecular mechanisms on diseases provided a surprising, sometimes contradictory, view of the autophagic effect.

## 4. Dual Role of Autophagy in Cancer Regulation 

About 25 years ago, researchers suggested that the deletion of ATGs in human cells might be related to tumor growth both in vitro and in vivo. Today, a vast number of investigations demonstrate that autophagy is overactivated, deregulated, or suppressed in cancer and that autophagy-related pathways crosstalk with tumor suppressor or oncogenes. However, it remains controversial whether autophagy works as a pro-death (anti-cancer) mechanism or a pro-survival pathway. The general overview of the role of autophagy in cancer is that this catabolic mechanism exerts cytoprotective effects, thus preventing tumor initiation. Nonetheless, once the primary tumor is established, autophagy represents an important mechanism by which cancerous cells take advantage to improve the proliferative rate and escape from anti-cancer treatments (Figure 2). A classic example may be found in breast cancer, in which autophagy acts as a tumor suppressor mechanism during the early stage since it degrades tumorigenic factors (such as protein aggregates, oncoproteins, and damaged mitochondria) and supports the immune response. Consistently, the autophagy-related gene signature results are frequently lost during the initiation of breast cancer [108]. Contrarily, autophagy helps the survival, invasion, and migration of breast cancer cells, thereby supporting the metastatic process and confirming the dual role of autophagy in cancer. Moreover, it is important to remember that, for classifying a malignancy, it is not possible to consider only (i) when the tumorigenesis starts and (ii) when the tumor is formed. Most cancers have at least five (0, I, II, III, and IV) broad stages [109]. It should not be surprising that autophagy may be differentially expressed among all these different stages. To make everything more difficult, tumors not only have different stages but also different subtypes, which have profound distinctions in the molecular expression characteristics and clinicopathological profiles. In line with this, breast cancer has four different subtypes [110]. Each subtype shows a different stage, and it has been demonstrated that none of them possess the same autophagy-related gene signature as well as a similar response to autophagy activators or inhibitors [111,112,113]. Unfortunately, specific investigations do not exist in which the autophagy role has been well associated with an exact subtype of a specific cancer type. Investigations are limited to verify the sensitiveness of cancer subtypes to a curative treatment with chemotherapy agents and autophagy modulators [112,114]. Not less important, all these studies have been performed in cancer cell lines and not in primary cells obtained by patients affected by different cancer subtypes. Opposingly, several investigations analyzed the autophagy-related gene signature in samples obtained from individuals affected by various cancer types, and, in certain cases, investigators also analyzed the autophagy levels in diverse cancer subtypes [115,116,117] (Table 1). However, the primary aim of these studies was to identify new factors that could help prognosis and therapy. They totally lack associating the level of expression of the autophagy gene investigated with the effective autophagy function and levels at the time of sample collection, thereby making it difficult to associate the autophagy with the cancer subtype. In the following sections, we elucidate the role of autophagy in cancer by providing evidence showing how autophagy supports or counteracts the cancer establishment and growth and its role during cancer therapy. Undoubtedly, it is crucial to well understand the dual role of autophagy in this context before making the modulation of autophagy as an effective approach to defeat cancer. 

### 4.1. Autophagy as a Tumor Suppressor Mechanism 

A balanced revision of the manuscripts published in the field agrees with the fact that autophagy orchestrates tumor suppressor activities in terms of (i) a preventive action against neoplastic transformation of healthy cells and (ii) at early stages of cancer development, though with some controversial data [127,128,129]. To achieve these conclusions, some considerations should be made; they indeed make the full understanding of autophagy in cancer difficult, and, with this, the ways by which it acts as a tumor suppressor mechanism. The first one is the high interconnection of autophagy with several different pathways (i.e., inflammation, oxidative stress, apoptosis, pro-survival signaling routes); second, the fact that autophagy-related genes regulate not only autophagy itself [130]; thus, it is not always possible to achieve unambiguous results; third, whole-body and homozygous knockout (KO) animal models for most of the genes involved in autophagy are lethal at birth. This means that, without mosaic model deletions, both the long-term and complete loss of the autophagic properties of a given gene are not currently reachable. 

In the 1990s, by analyzing human breast carcinoma cell lines, deletions of the key autophagy protein BECN1 were identified in 9 out of the 22 cell lines analyzed [131]; this research led to consider and investigate in depth the autophagy process as a putative key step in tumor suppression. In 1999, this evidence was translated into in vitro systems and animal models, where, for the first time, the group led by Levine B. reported proof for the role of BECN1-dependent autophagy in oncogenesis. Indeed, BECN1 deletion, accompanied by impairment in autophagy, favored the onset of cell proliferation and malignances [132]. Additional research reported BECN1^+/−^ mice with 2-fold increased spontaneous tumorigenesis and significant proliferative lesions compared to BECN1^+/+^ littermates and a related increased risk of mortality [133,134]. This outcome was dependent on the absence of the BECN1 gene and not on mutations in the wild type remaining one. In agreement, its reintroduction significantly inhibited tumor progression [132].

Recently, the primary role of BECN1 in tumor suppression as a positive regulator of autophagy was partially hidden by the discovery of alternative pathways in which it plays regulatory roles. This is the case of the wingless-type MMTV integration site family member 1 (WNT1)-driven breast oncogenesis [135] in which the haploinsufficiency of BECN1 upregulates WNT1 signaling and TNF receptor superfamily member 11a (TNFRSF11A)—nuclear factor kappa-light-chain-enhancer of activated B cells (NF-kB) pathway to sustain tumor growth, and that was mediated by the cadherin-catenin complex [136]. Herein, BECN1 deletion would avoid the membrane localization of E-cadherin, a protein with tumor suppressor properties, favoring the transcription of genes involved in tumor progression [137]. This feature is selective for BECN1 and is lacking for other ATGs. Importantly, the absence of E-cadherin in membrane compartments was annotated in several cases of breast cancers where the prognosis was poor [138,139]. Endophilin B1 (BIF1) is an interactor of BECN1 and is known to induce autophagy by enhancing autophagosome formation after starvation [140]. The interaction of BECN1/BIF1 in triggering the autophagy flux is essential to minimize events of tumorigenesis. Indeed, this process, which becomes impaired in BIF1^−/−^, mice, leads to spontaneous tumor incidence, preferentially lymphoma but also several solid cancers [140].

As for BECN1, the deletion of other autophagy genes also suggested a role in proliferative lesions, although presenting the onset of primarily benign masses. For example, ATG5 whole-body suppression in mice, for short periods, was predisposed to cell proliferation [141]; this result confirmed what was reported several years earlier, where the evaluation of ATG5 deletion in all tissues of a mouse model led to the identification of benign tumors exclusively localized in the liver [142]. As observed for ATG5, ATG7^−/−^ mice also preferentially exhibit liver oncogenesis [142]. These patterns extensively differ from those regarding BECN1, where tumor onset was not limited to a given organ but involved several tissues and with worse prognosis. This difference may validate the hypothesis that, in the case of BECN1, the resulting phenotype may not be completely ascribed to autophagy impairment. 

As reported above, AMBRA1 is a known regulator of autophagy. AMBRA1 deficiency in mouse models increases the risk to develop melanoma, significantly enhances melanoma onset, cell proliferation, invasion, and metastasis compared to AMBRA1 WT [143,144]. Matching the roles of AMBRA1, between the modulation of autophagy and the genesis of malignances, is challenging. Indeed, autophagy disruption by AMBRA1 deficiency in melanoma cells was found to be compensated by alternative mechanisms, shedding light on new routes regarding its involvement in cancer. As demonstrated, AMBRA deficiency has been correlated to the hyperactivation of the focal adhesion kinase 1 (FAK1) signaling, that, in turn, it is fundamental to provoke cell invasion and melanoma growth [145]. Interestingly, all these phenomena were not related to the pro-autophagy functions of AMBRA, thus suggesting the existence of compensatory events and a new therapeutic strategy proposing FAK1 inhibition for AMBRA1 low-expressing melanoma [143].

However, which are the mechanisms responsible in autophagy-mediated tumor suppression? Autophagy limits the onset of events that facilitate oncogenesis, such as genomic instability, inflammation, and ROS production, but, also, in the healthy cell, it plays a crucial role as a mechanism of detoxification to prevent the stasis of molecules, pathogens, and proteins that contribute to the neoplastic insult.

In 2007, autophagy was proposed as a mechanism crucial for genomic maintenance [146], a hallmark of almost all tumor types. In detail, autophagy has been linked to the DNA damage response [147], DNA repair mechanisms [148], and replication stress [149]. Indeed, autophagy can be modulated by the DNA damage response, and, in turn, this can be modulated by autophagy. This involves major mechanisms, including the activation of the serine/threonine kinase ataxia telangiectasia mutated (ATM) that, by triggering the cascade mediated by the AMPK/TSC2/ULK1 axis, induces autophagy in response to genotoxic stressors safeguarding the cell [150]. Another sensing damaging factor that links autophagy to DNA damage is the tumor suppressor p53. Following genotoxic stress, p53 induces the transcription and activation of the damage-regulated autophagy modulator (DRAM), a lysosomal protein that, by inducing autophagy, accompanies p53-mediated cell death [151]. Accordingly, clinical investigations about DRAM expression in different kinds of tumors reported a strong downregulation in its mRNA and protein in pathological tissues [151]. Moreover, sestrins (SESN1/2) participate in the downstream network of p53 in the inhibition of cell proliferation and tumor growth [152]. The link between SESN1/2, cell death, and autophagy relies on the fact that SESN1/2 inhibits the MTOR pathway by promoting the activation of the TSC1/2 complex and AMPK [152]. Last but not least, the enzyme PARP1, which senses single strand breaks and, as a consequence, is upregulated, guarantees the induction of autophagy via AMPK [153], maintaining low levels of ATP and high levels of AMP.

The detoxification role of autophagy is ascribed to its ability in the successful degradation of harmful molecules and proteins and dysfunctional organelles. Among the oncogenic proteins being removed by autophagy, we found MYC kept at low levels through a fine regulation of cellular inhibitor of PP2A (CIP2A) expression, responsible for MYC dephosphorylation, a signal for its proteasomal degradation [154]; the list goes on with mutant p53 [155], hexokinase 2 (HK2) [156], and translationally controlled tumor protein (Tpt1/TCTP) [157]. These findings, together with the knowledge that all are proteins overexpressed in cancer and that their autophagy-mediated removal inhibits tumor progression, confer once again to autophagy defensive mechanisms against either cell transformation or cancer development at early stages.

As reported above, the most common mechanism of organelle degradation by autophagy is mitophagy [158]. Interestingly, one of the proteins involved in mitophagy, BNIP3, was found deleted in breast cancers, especially in the triple-negative phenotype. Routinely, BNIP3 modulates mitophagy to remove dysfunctional mitochondria; in animal models of breast cancers, the absence of this protein favors the increase of ROS production, which favors the oncogenic effect of hypoxia inducible factor 1 subunit alpha (HIF1α) under hypoxia in terms of increased angiogenesis and malignances progression [159]. ROS generation derives from hypoxic conditions and deprivation of nutrients; in a noteworthy manner, they own the properties to directly target ATG proteins, such as ATG4, causing its oxidation; as a consequence, this prompts a burst of autophagy [160]. Autophagy is also involved in the quick fix of pathological ROS using high mobility group box 1 (HMGB1) as the main mediator. The shuttle of HMGB1 from the nucleus to the cytoplasm, and then its release in the extracellular milieu, enhances LC3-II puncta formation and promotes p62 degradation and autophagolysosomal formation [161]. HMGB1 is an interactor of BECN1 and promotes the disruption of BCL2/BECN1 binding via a direct competition [162]. This disrupted interaction prompts autophagy. Several effectors play this role in disrupting the BCL2/BECN1 binding; as well, JNK1, which becomes active upon oxidation, follows the same pattern [163]. This has a great impact on cancer, modulating autophagy and cell death pathways at the crosstalk with the BCL2-like proteins. In summary, despite the vast majority of the tumors’ type identified as dependent by autophagy impairment being benign, this suggests that autophagy becomes essential for the progression of cancer further in late stages.

### 4.2. Autophagy Drives Tumor Growth of Established Tumors, Metastasis, and Resistance to Therapy 

Increased autophagic activities have been reported in different cancer types, such as melanoma and breast cancer, in which the increased autophagy flux was also associated with a more aggressive phenotype [164,165]. Increased autophagy also helps cancer cells to survive in harmful conditions, such as nutrient privation or hypoxia. Notably, cancer cells use glycolysis to metabolize lactate. It has been observed that, in liver cancer, the levels of HK2, a rate-limiting enzyme of glycolysis, is kept low by the autophagic process [166]. In addition, lactate secretion increases the extracellular pH, an event that can activate autophagy [167]. Consistently, the increased expression of autophagic markers have been found in acute acidification of breast cancer cells, and, meanwhile, a low pH reduced proliferation rate. These data suggest that cells exposed to low pH use autophagy as a pro-survival mechanism [168]. Autophagy also helps tumor cells to grow under conditions of hypoxia. The tumor mass lacks an efficient vasculature during the uncontrolled cell proliferation. Inside the mass, it creates hypoxic (oxygen < 0.3%) and (anoxic < 0.1%) areas that activate the family protein members HIFs, in particular the isoform HIF-1α, which activates BNIP3L and BNIP3 [169]. These factors are then able to induce autophagy by disrupting the BCL2-BECN1 complex and, finally, avoid the cell death activation. Furthermore, HIFs are implicated in regulating the metabolism. Indeed, once activated, they upregulate the activity of glycolytic enzymes, thereby promoting increases of the anaerobic metabolic flux. The elimination of mitochondria by autophagy is a fundamental event during carcinogenesis since it could help to reduce the ROS production and increase the nutrient and oxygen availability, thus promoting the cellular survival. Different notions support the possibility that mitophagy is a protective mechanism used by cancer cells to escape from cell death and identified alteration of the activity of numerous mitophagy partners (such as PINK1, Parkin, BNIP3L, NIX, and FUNDC1) in several cancer types, including glioblastoma, breast cancer, hepatocellular carcinoma, acute myeloid leukemia, and prostate cancer. However, depending on the cancer type, the same protein may have both tumor suppression or promotion properties. For example, this happens for the mitophagy regulator BNIP3, which exerts tumor promotion in melanoma and pancreatic cancer; meanwhile, it is protective against breast cancer. Further studies are needed to unveil the exact role of these mitophagy regulators during carcinogenesis. Tumor growth and the spread of cancer cells in organ/tissues different from the site of origin need adequate levels of oxygenation and nutrients. Autophagy contributes to develop new blood and lymphatic vessels to respond to this high energetic demand. Indeed, ATG5 was found to promote angiogenesis by regulating the activities of HMGB1, which, in turn, activates autophagy by binding BECN1. The axis HMGB1–BECN1 was also found to improve the tumor growth in leukemia [170] and was found to be a fundamental regulatory mechanism in developing mesothelioma and other asbestos-related malignancies [171]. In this case, it has been recently demonstrated that, during asbestos exposure, cytoplasmic and extracellular HMGB1 led to the transformation of mesothelial cells by activating autophagy through the RAGE–MTOR–ULK and BECN1 pathways [171]. For cancer cells, it is fundamental to detach from the tumor of origin to proliferate in a “new and foreign” microenvironment. At the same time, cancer cells detached must evade to anoikis, a particular form of apoptosis that occurs upon cell detachment [172]. Autophagy helps cancer cells to do this. Autophagy favors the survival of metastatic cells during detachment and is activated during anoikis [173]. Furthermore, ECM elements activate autophagy [174]. Not surprisingly, by blocking the autophagy process, the matrix detachment of metastatic cells diminishes, and suppressed autophagy is associated with reduced levels of pro-invasive cytokines [175].

Autophagy also has a prominent role in the survival and maintenance of cancer stem cells (CSCs), a small subpopulation of cancer cells that possess high metastatic potential, extraordinary self-renewal, great aptitude to be a quiescent cell pool, and high differentiation capabilities, all features that are responsible for tumor initiation, growth, invasion, and recurrence. First identified in acute myeloid leukemia [176,177], CSCs were subsequently found in almost all solid cancers, such as lung [178], melanoma [179], brain [180], pancreatic [181], and breast [182] cancers. From then, different investigations have been performed to understand which cellular molecular mechanisms are important for the growth and survival of cancerous cells. Recent works identified autophagy as a crucial pathway. Indeed, upregulated levels of BECN1 and ATG14 and of other autophagic partners (such as ATG4, ATG5, ATG12, and LC3) were identified in breast CSCs [183,184,185,186] and correlated to survival, expansion, and therapy resistance of CSCs. The interconnection between autophagy and CSCs has also been identified in other malignancies. Liver CSCs required high autophagy to survive under oxygen- and nutrient-deprived conditions [187]. Similar findings were also observed in pancreatic ductal adenocarcinoma [188] and osteosarcoma (OS) CSCs, in which the higher autophagy levels found in these cells were essential to give tolerance against low nutrients and hypoxia [189]. Consistent with this, autophagy-deficient OS CSCs lose the advantage of resistance to these unfavorable conditions. Sustained activation of autophagy has also been found in colon CSCs. In this case, autophagy not only is essential for the maintenance of CSCs but also gives chemoresistance via activation of the gene caudal type homeobox 1 (CDX1), which stimulates proliferation and tumorigenesis regulating the molecular network composed of p53 and BCL2 [190]. Autophagy represents a key element inducing resistance during cancer therapy. Indeed, target therapy, chemotherapy, and radiotherapy activate cytoprotective autophagy [191]. For example, the chemodrugs tamoxifen and bortezomid used to treat breast cancer induce autophagy through the UPR-signaling pathway and MTOR inhibition [192,193]. Bevacizumab is a recombinant monoclonal antibody approved treatment for colorectal cancer. Bevacizumab was found to increase the number of autophagic vesicles and autophagic markers to protect cells from cell death. Consistent with this, autophagy inhibition restores the effect of bevacizumab to kill cancerous cells in both in vitro and in vivo experimental studies [194,195]. Similar evidence was also found for apatinib (another compound used for CRC), which induces protective autophagy by interfering with MTOR activities and stimulating ER stress through inositol-requiring enzyme 1 (IRE1) signaling pathway. Contrarily, apatinib, in combination with autophagy blockers, induces cell death in CRC cell lines and in xenografts in nude mice [196,197]. Currently, imanitib is used to treat chronic myeloid leukemia (CML). Recent studies unveil that imatinib reduces the levels of miR-30, which is a negative regulator for BECN1 and ATG5 [198]. The fact that the downregulation of the expression of these autophagic elements increases autophagy suggests a protective role of imanitib-induced autophagy BECN1 and ATG5 [199]. Similar autophagy-dependent protective effects were also found for ponatinib, perifosine (against CML) [200,201], and dasatinib (against chronic lymphoid leukemia) [202]. Melanoma represents a main cause of cancer-related death worldwide. A primary targeted therapy approved for this cancer is to block the BRAF activity with specific inhibitors, such as vemurafenib, dabrafenib, and encorafenib. It was demonstrated that BRAF inhibition activates the autophagy pathway, increases TGF-β levels, and boosts the EMT machinery, provoking tumor proliferation and resistance to the chemotherapy [203]. Undoubtedly, radiotherapy is one of the key treatments for several cancer types and allows improving the survival time and life quality of patients. However, in thyroid cancer, ion radiations induce ROS generation that mediates the activation of PI3K–Akt-dependent protective autophagy [204]. The induction of autophagy was also observed in breast cancer treated with irradiation, suggesting a protective effect of autophagy promoted by irradiation [205]. Similar observations were also achieved in esophageal, pharyngeal, and cervical carcinoma cells [206,207]. In all of these, the excessive increase of autophagic vesicles and different autophagy markers detected after irradiation were associated with an increase of the proliferation rate and survival of cancerous cells.

## 5. Autophagy and Tumor Immune Response

It is widely accepted that the human body, against a tumor, produces an innate immune response, in which immune cells (such as natural killer, NK, macrophages, and T cells) and immune effector molecules (such as CD8+ and CD4+) are the main effectors [208]. Improving the immune function of these elements with specific immunotherapeutic agents has shown great efficacy against cancer. Consistently, several immunotherapy agents have been approved by the FDA to enhance the immune response and kill cancerous cells, including cytotoxic T cells (such as CD19-targeting chimeric antigen receptor, CAR, T cells) [209,210] and inhibitors of immune checkpoints that target cytotoxic T lymphocyte (CTL)–associated protein 4 (CTLA4), PD-1, and its ligand, PD-L1 [211]. In addition, a series of compounds exist that look to be efficacious in increasing the immune response in pre-clinical investigations or that are present in active investigations in clinical trials. Among them, the most promising are cancer vaccines, macrophage inhibitors, and NK-targeted therapies [212].

Autophagy has an important impact on tumor immunity. Autophagy can degrade the effector immune molecules of NK and CTL [213,214]. Autophagy can also regulate the expression and stability of the PD-L1 protein [215,216] and the immunogenic cell death of tumor cells by controlling the release of damage-associated molecular patterns (DAMPs) [217], which are chemical attractants for dendritic cell (DC) precursors. Autophagy is also important for the T cell biology: it supports the activation, proliferation, function, and memory maintenance of these cells [218,219]. Similarly, autophagy is fundamental for B cells since a lack of autophagy attenuates the secondary immune response, antigen presenting and processing, and the differentiation of plasma cells [220]. Finally, autophagy controls the functions of dendritic cells, the activity of myeloid derived suppressor cells (MDSCs), and regulates the formation, maintenance, recruitment, and polarization of macrophages [208]. Considering all these findings, it is not surprising that autophagy can impact cancer immunotherapy. Consistently, the inhibition of autophagy with lysosomotropic agents augments the TRP2 peptide vaccination-mediated anti-tumor immunity in melanoma tumors by restoring the CTL-mediated lysis [221]. High-dose interleukin 2 (HDIL-2) administration is an FDA-approved treatment. However, this treatment causes several side effects that limit its use. Interestingly, this toxicity has been associated with an HDIL-2-dependent activation of autophagy, and the inhibition of autophagy decreases the toxicity and enhanced immune cells proliferation and infiltration, thus providing an alternative therapeutic approach for the treatment of patients with renal cancer and hepatic cancer and melanoma [222]. The inhibition of autophagy also demonstrated efficaciousness in improving the antitumor efficacy of IL-24 in oral squamous cell carcinoma [223] and to overcome the resistance of cancer cells to cytotoxicity induced by the cytokines IFNγ and TNF [224].

## 6. Autophagy Modulation for Cancer

In vitro and in vivo studies confirm that the modulation of autophagy represents a promising opportunity for cancer therapy. Considering the dual role of autophagy during the different phases of a cancer, it is not surprising that activators as well as inhibitors are both feasible anti-cancer compounds that might improve the efficacy of chemotherapeutic agents. Hence, we provide an overview of the key examples (Figure 3). 

### 6.1. Autophagy Stimulation for Cancer Treatment

#### 6.1.1. MTOR Inhibitors

As reported above, MTOR represents the main repressor of the autophagic machinery. Rapamycin and analogues, known as rapalogs (such as everolimus and temsirolimus), are selective inhibitors of MTORC1, and they have demonstrated to block the tumor growth and induce cell death in lung cancer, breast cancer, neuroblastoma, hepatocarcinoma, and osteosarcoma [225]. Conversely, some of them were also found to stimulate autophagy and cell proliferation in the cancer context. This is probably due to rapalogs’ limited efficacy to block the pro-survival and proliferative activities of MTORC2 and other signaling pathways related to the cellular growth [226]. Considering these aspects, rapalogs seem to be suitable for combined anti-cancer therapies that permit to interfere with diverse intracellular molecular pathways. Consistent with this, newly developed inhibitors of both MTOR complexes determine MTOR inhibition with the consequent activation of autophagy cell death and cell cycle arrest and apoptosis. 

#### 6.1.2. Histone Deacetylase Inhibitors

Inhibitors of histone deacetylases (HDACs) demonstrated competent to induce cell death in different cancer types. Consistently, the HDAC inhibitors romidepsin and vorinostat are FDA approved compounds for the treatment of T-cell lymphoma. Interestingly, vorinostat also mediates cell death by activating autophagy via the lysosomal cysteine protease cathepsin B [227]. Similarly, another HDAC inhibitor, MHY2256, induces apoptotic and autophagic cell death in in vitro and in vivo models of endometrial cancer regulating the p53 acetylation [228]. Interestingly, it has been demonstrated that autophagy is also critical for the acquired resistance for HDAC inhibitors therapy. Indeed, vorinostat-induced autophagy was found to switch from a cell death promoting signal to a cytoprotective mechanism to protect cells from anti-cancer therapies. Consistent with this, by using inducers (such as rapamycin) as well as inhibitors (like chloroquine) of the autophagic pathway, the efficacy of vorinostat to promote cell death was synergized or restored, respectively [229]. Improving the knowledge of the mechanism of HDAC inhibitors in cancer and their role in modulating autophagy would facilitate the clinical use of these compounds and unveil new suitable combinations. 

#### 6.1.3. BH3 (BCL2 Homology 3) Mimetics

These agents represent a group of molecules that mimic the interactions of BH3-only proteins and activate autophagy by disrupting the inhibitory interaction of BCL2 family members with BECN1. In prostate cancer, the BH3-mimetic sabutoclax blocks the prosurvival BCL2 family member, myeloid cell leukemia-1 (MCL1), and activates the autophagic process to facilitate the induction of apoptosis mediated by NOXA and BIM [230]. Similar effects were also found for the other BH3 mimetic compound obatoclax. Indeed, in acute lymphoblastic leukemia (ALL), obatoclax provokes disruption of the interaction between MCL1 and BECN1, inactivation of MTOR, and activation of the necroptotic cell death [231]. Gossypol, a natural BCL2 homology domain 3 mimetic compound, antagonizes the inhibition of autophagy by BCL2 and induces autophagy cell death in different cancer types, such as mesothelioma, neck cancer, and colon cancer. However, the specificity of this compound to determine the reactivation of autophagy by disrupting the interaction between BCL2 and BECN1 remains doubtful. Indeed, gossypol was found to target the RNA-binding protein musashi-1 (MSI1) (that works as a regulator of mRNAs) [232], modulate the expression and functioning of p53, ERBB2, p38, and AKT [233], and, most importantly, also act in an BECN1-independent manner [234]. 

#### 6.1.4. Tyrosine Kinase Inhibitors

Tyrosine kinase (TK) are enzymes that phosphorylate tyrosine residues of specific proteins. TK are deeply involved in regulating the cell proliferation and survival in different cancers. Interestingly, the use of TK inhibitors (TKI) interferes with cancer growth [235], and, recently, this characteristic has been linked to the modulation of the autophagic process. For example, sorafenib is the main chemotherapy used for hepatocellular carcinoma, and it was demonstrated to switch autophagy from a cytoprotective role to a death-promoting process via the regulation of AKT activity [236]. The same effects were also found in renal cancer cells [237]; meanwhile, in prostate cancer, sorafenib activates autophagy to promote the necroptotic cell death [238]. Moreover, erlotinib induces autophagy to induce the cell death of cancerous cells. This was observed in non-small cell lung cancer in which the exposure of erlotinib provokes the increased expression of autophagic markers (such as LC3 and ATG5/7), suppression of MTOR signaling, and activation of p53 [239]. Unfortunately, one main limit of TKI is to create resistance to cell death. It has been demonstrated that combined treatment of TKI-resistant cancer cells with autophagic inhibitors may overcome the TKI-dependent cell death resistance [240,241]. 

### 6.2. Autophagy Inhibition for Cancer Treatment

#### 6.2.1. ATG Inhibitors

ATGs are fundamental proteins for the autophagy initiation and formation of the autophagosome vesicle. Interfering with their activities represents a promising approach to counteract the excessive autophagic levels of cells. ATG4 is responsible to process LC3 in the cleaved form conjugated to PE. NSC185058 blocks the activity of ATG4, reduces the number and size of autophagosome, and has no effect on the activities of other kinases, such as MTOR. Furthermore, NSC185058 reduces osteosarcoma growth in vivo [242]. Moreover, the compound UAMC-2526 displays an inhibitory effect toward ATG4. Interestingly, UAMC-2526 abolishes the autophagic process in mice bearing colorectal tumors and increases the efficacy of chemotherapy to provoke cell death and reduces the tumor growth [243]. Recently, in silico screening and in vitro assays identified another important ATG4B inhibitor (S130), which demonstrated the ability to interfere with the proteolytic activity of ATG4 and not with other proteases. Importantly, S130 distributes well in tissues in in vivo models, induces the cell death of colorectal cancer cells, and reduces the tumor size [244]. Therefore, these studies identify ATG4B as a potential anticancer target.

#### 6.2.2. ULK1 Inhibitors

Together with MTOR and AMPK, ULK1 is one of the major regulators of autophagy. Not surprisingly, ULK1 results upregulated in several human diseases, including cancer, and counteracting its activity seems to be a promising anti-cancer treatment. Consistently, the ULK1 inhibitor SBI-0206965 determines tumor growth arrest in numerous cancers, such as neuroblastoma, renal carcinoma, and NSCLC [245,246]. Interestingly, apart from its inhibitory effect toward ULK1, SBI-0206965 also blocks the activity of AMPK [247], and, when combined with MTOR, the inhibitors greatly increase the cell death of tumor cells [248]. ULK-100 and ULK-101 are described as potent molecules suppressing autophagy and ULK1. ULK-101, which displays superior selectivity to ULK-100, suppresses the induction of early autophagy and reduces the cell viability of NSCLC [249]. Besides these molecules, other molecules were identified as ULK1 inhibitors. By screening a collection of 764 compounds against ULK1, a crystallography study identified six potential ULK1 inhibitors [250]. Further studies are necessary to verify this characteristic in vitro as well as and in vivo. Another screening study demonstrated that MRT67307 and MRT68921 block ULK1 activities [251]. This study was performed in vitro, and subsequent investigations demonstrated that these compounds are efficacious in reducing the tumor growth [252,253]. Overall, the inhibition of ULK1 seems to be a promising strategy for cancer therapy. However, all these compounds have several off-target liabilities. More knowledge of their molecular mechanism of action is necessary. 

#### 6.2.3. PI3K Inhibitors

The phosphoinositide 3-kinases (PI3Ks) family presents three different classes with different functioning. Meanwhile, the role of class II regarding the autophagic context remains unclear; class I is responsible for the inhibition of autophagy through MTOR activation by the PI3K/AKT pathway. On the contrary, class III (VPS34) activates autophagy. 3-Methyladenine (3MA) is the most common PI3KCI inhibitor, and several reports demonstrate that 3MA improves the effects of chemotherapy drugs. Unfortunately, 3MA exerts its functions at high concentrations, making it difficult to use in vivo. Furthermore, 3MA presents a dual role on autophagy. Indeed, under caloric restriction, it induces autophagy by inhibiting the PI3KC1. Other PI3KC1 inhibitors widely used in research are wortmannin and LY294002. The first one was found efficacious to reduce the cell viability and increase the apoptosis in breast cancer and in TRAIL-resistant colon cancer cells [254]. LY294002 demonstrates the ability to reduce autophagy to enhance the cytotoxic effect of temozolomide in melanoma cells and of 5-fluorouracil (5-FU) against esophageal squamous cell carcinoma [255,256]. However, LY294002 has low solubility and pharmacokinetics. The development of LY294002 analogs, such as SF1126, seems to overcome this issue. Indeed, SF1126 accumulates in tumor sites and inhibits the growth of colorectal cancer cells also in in vivo models [257]. In addition to its inhibitor effect against PI3KCI, SF1126 downregulates MYC and cyclin D1 proteins and activates the p38 signaling [257]. 

#### 6.2.4. Lysosome Inhibitors

Probably, the use of compounds blocking the fusion of autophagosome with lysosomes (lysosomal inhibitors) represents the best pharmacologic approach to provoke the inhibition of autophagy and obtain anti-cancer effects or potentiate the efficacy of antineoplastic drugs. Among them, the most used compounds are chloroquine (CQ) and its derivatives. CQ and analogs are weak bases that move through the cell membranes and enter into lysosome as unprotonated forms. In this regard, thanks to the high H^+^ concentration, CQ and derivatives are protonated and trapped in the lysosome. In this conformation, they cause increasing lysosomal volume and block the activity of lysosomal enzymes. These compounds are the promising drug candidates for the fight against cancer. Indeed, CQ and hydroxychloroquine (HCQ) are the only lysosomal inhibitors approved for cancer clinical trials, and their efficacy toward cancer will be discussed in the following section. Apart from CQ and HCQ, other CQ-derivatives exist that seem to have important effects for cancer treatments. Superior potency compared to CQ was found for the quinolone analog mefloquine (MQ), which has been found to produce anticancer effects in various breast cancers [258], in prostate cancer cells [259], and in chronic myeloid leukemia (CML) [260]. Interestingly, in this latter investigation, it was proven that MQ preferentially targets CML CD34+ stem/progenitor cells and potentiates the anti-cancer effects of tyrosine kinase inhibitors [260]. Unfortunately, the efficacy of MQ has not been tested yet in in vivo models. Lys05 and its soluble version Lys01 are novel lysosomal autophagy inhibitors and are more potent than CQ and HCQ. Furthermore, Lys05, used at a lower dose, also possess antitumor efficacy in mice with human colorectal cancer xenografts [261], thus demonstrating a viable compound for anticancer therapy based on autophagic inhibition. The dimeric quinacrine DQ661 was identified by performing a screen of novel dimeric antimalarial agents and demonstrated to have a superior anticancer effect compared to other antimalarials (such as MQ, primaquine, and quinacrine) toward melanoma, colorectal, and pancreatic tumors. Furthermore, apart from the anti-autophagic effect, DQ661 blocks the activity of palmitoyl-protein thioesterase 1 (PPT1), resulting in the disruption of the MTORC1–RHEB interactions and, lastly, impairment in MTOR signaling [262]. There are, thus, other molecules that can disrupt the lysosomal functioning. Some of them are natural products, such as the polysaccharide ganoderma lucidum (GL), from the fungus *Ganoderma lucidium*, that, by targeting lysosomes, inhibit autophagy and induce apoptosis of prostate [263] and colorectal cancer cells [264]. Other molecules are chemical compounds that disrupt the lysosome functioning, traffic, and formation of autolysosomes. These may be represented by a family (termed as WX8-family) of five inhibitors of the FYVE finger-containing phosphoinositide kinase (PIKFYVE) that are a hundred times more potent than CQ and HCQ and selectively kill autophagy-dependent melanoma cells without having an effect on the non-cancerous cells [265]. Finally, special attention should be paid regarding the possibility to repurpose in oncology a series of drugs used for managing other human diseases. This is especially true for the FDA-approved drugs for the treatment of psychiatric and anti-depressant disorders. Despite the exact molecular mechanism remaining not clearly unveiled, growing evidence supports that these compounds impair autophagy flux by blocking the lysosomal degradation. Similarly, different investigations also demonstrate that these drugs have anti-cancer properties. Only to name a few, clomipramine (CM) and its metabolite desmethylclomipramine (DCMI) have been determined to decrease the cell viability of cancer stem cells (CSCs) [266]. Fluoxetine has anti-proliferative effects against breast cancer [267] and increases the efficacy of paclitaxel in gastric cancer [268]. Clozapine promotes the cell death of non-small cell lung carcinoma cell lines [269]. 

## 7. Cancer Clinical Trials 

As reported above, the most promising autophagy modulators against cancer are the autophagy inhibitors CQ and HCQ. The addition of these molecules potentiates the effect of chemotherapy and radiotherapy in several cancer types, both in vitro and in vivo. For example, CQ augments the cytotoxicity of the chemotherapy treatment in tumors with the tumor suppressor PML absent or downregulated [270,271]. Notably, PML depletion promotes autophagy activation [270]. Colon cancer cells acquire resistance to the chemotherapy agent bevacizumab and oxyplatin via autophagy activation [272]. In vivo experiments demonstrate that CQ overcomes this resistance and sensitizes colon cancer cells to chemotherapy [272]. Lysosomotropic agents are also demonstrated to be effective toward melanoma. In this case, the HCQ synergizes with the MTOR inhibitor temsirolimus to suppress melanoma growth [273]. Similarly, HCQ combined with tamoxifen or gefitinib increased the cell death of breast cancer cells [274,275]. Notably, one main feature of CSCs is to be resistant to conventional anticancer therapy and to determine long-term tumor remission. Several experiments suggest that CQ and HCQ are also useful to reduce these characteristics of CSCs. For example, the viability of gastric CSCs isolated from gastric cancer cell lines is reduced when the chemotherapy agent 5-FU is used in association with CQ [276]. The combinatory treatment of CQ with doxorubicin and docetaxel is efficacious to reduce the stemness of breast CSCs and increase the sensitivity to chemotherapy compounds [277]. Furthermore, this work demonstrates that, by using nanoparticle systems to deliver chemotherapy agents and CQ, it is possible to improve the biodistribution of both drugs, suppress the tumor growth and size, and reduce the CSCs population in vivo. The CQ-mediated inhibition of autophagy also helps to overcome the cisplatin resistance in bladder cancer cells. In this case, the co-treatment of CQ and gemcitabine and mitomycin induces cell death and reduces the stemness of cisplatin-resistant bladder cancer cells by regulating the JAK/STAT pathway [278]. Similarly, autophagy inhibitors enhance the anti-cancer effects of the standard first line drug imatinib mesylate (IM) for patients with chronic myeloid leukemia (CML) [279]. IM treatment in combination with CQ or other anti-autophagic compounds also determines a selective elimination of CML stem cells, thus providing an alternative therapeutic approach to target the stem cell CML populations that are IM-resistant [279]. Along similar lines, the combination of CQ with salinomycin interferes with the proliferation and survival of breast CSCs [280]. CQ is also efficacious to improve cancer immunotherapy. Consistently, the inhibition of autophagy with this lysosomoptropic agent augments the TRP2 peptide vaccination-mediated anti-tumor immunity in melanoma tumors by restoring the CTL-mediated lysis [221]. Similarly, CQ combined with HDIL-2 (used for the treatment of melanoma and renal carcinoma) improves both the infiltration and proliferation of immune cells, enhances anti-tumor effects, and expands the survival time in a murine liver metastasis model [222]. Altogether, these preclinical investigations have permitted to develop the rationale to use CQ and HCQ in clinical trials: CQ has been included in 22 clinical trials and HCQ in 77 (https://clinicaltrials.gov/, accessed on 30 October 2021). HCQ treatment remains preferred to CQ since the evidence demonstrated that HCQ is threefold less toxic than CQ [281]. In only three phase I/II trials for solid tumors and leukemia, HCQ is used alone. In all the other clinical trials, CQ or HCQ are used in combination with conventional chemotherapy agents (such as bortezomib, temozolomide, erlotinib, and gemcitabine) for the cure of myeloma, melanoma, and other solid tumors, including colorectal and breast cancer, or with radiotherapy for the battle against glioblastoma. There are, therefore, clinical trials for advanced solid tumors in which the anti-autophagic compounds CQ and HCQ are used in combination with MTOR inhibitors (such as temsirolimus or rapamycin) or other positive regulators of autophagy, in particular the FDA-approved HDAC inhibitor vorinostat. At first sight, the concomitant use of an inhibitor and an enhancer of autophagy may appear as a paradox. However, several studies demonstrate that certain cancer treatments activate cytoprotective effects and help cancer cells to survive. By using the anti-autophagic compounds, it is thus possible to overcome this resistance. The early results obtained by the clinical trials conducted give a widespread picture in which the efficacy of autophagic inhibitors to increase the efficacy of conventional therapies in different cancer types is extremely variable. In certain cases, the combinations improve the overall survival rate and reduce the tumor growth, while, in others, the efficacy registered is negligible or moderate. For example, a double-blind, placebo controlled clinical trial conducted on glioblastoma patients demonstrated that the co-treatment of CQ and radiotherapy or chemotherapy improved the survival rate and reduced brain metastasis [282]. In another study, the potential benefits of CQ co-therapy with bertezomid were unveiled in a phase I study performed in patients with relapsed or refractory myeloma. Indeed, of the 22 evaluable patients, 6 presented immediate progression, 10 had stable disease, 3 presented a very good response, and 3 had a minor response [283]. What is more, clinical trials performed with HCQ in combination with gemcitabine [284] or nab-paclitaxel and gemcitabine [285] described potential anti-cancer effects in patients with resectable pancreatic adenocarcinoma. Unfortunately, CQ and HCQ were not always found to improve the efficacy of chemotherapy. Negligible results were obtained in a phase I clinical trial of HCQ with erlotinib in patients with non-small-cell lung cancer with survival not increasing. Similarly, the data obtained from a phase I/II trial of HCQ with radiation and the adjuvant temozolomide highlight no significant increases in the survival rates in patients with glioblastoma multiforme [286] (Table 2). Considering all these facts, larger clinical trials need to occur to have more detailed information about the efficacy to use autophagy inhibition in cancer-affected persons.

## 8. Summary and Future Directions

The term autophagy was coined in 1963 at the Ciba Foundation symposium on lysosomes by Christian de Duve. From this year to the late 1980s, studies about autophagy were based on morphological analyses. From the 1990s began the “molecular era”, which culminated in 2016 when Yoshinori Ohsumi was awarded the Nobel Prize in Medicine for his contributions in elucidating the genetic basis for autophagy. During all these years, a wide range of research groups contributed to investigate the important role of the autophagy pathways in cells, from yeast to humans. Today, autophagy is considered a primary cellular process regulating cell survival, proliferation, metabolism, and development. Hence, it is not surprising that autophagy actively participates during the initiation and progression of all human diseases. In cancer, autophagy seems to have opposite roles, since it can limit the earliest stages of tumorigenesis, but also favors tumor progression. In this context, autophagy protects cancer cells from environmental stress and from chemotherapy, helps cancer cells to proliferate, and facilitates the metastatic process. Thus, autophagy seems to represent an optimal target for the battle against cancer, and, indeed, several clinical trials have evaluated (and are evaluating) the modulation of autophagy for cancer treatment. The results obtained are very promising: in certain cases, autophagy modulation blocks the tumor proliferation and potentiates the ability of conventional chemotherapy to kill cells. However, in other cases, autophagy presents limited or absent effects, probably due to a still marginal knowledge of several aspects of autophagy during carcinogenesis. To overcome these limitations, we need to answer some important questions; in particular: how is the autophagic activity differentially regulated in different cancers? Is it better to target autophagy at an early or late stage to maximize the benefit? How do we develop rationally based therapeutic interventions based on autophagy modulation? Do autophagy modulators exist that are more potent, more selective, and with limited adverse effects that can improve the clinical effects observed with CQ and HCQ? Only a better understanding of the molecular mechanism that autophagy elicits during the different stages of carcinogenesis will help to answer these important questions and develop efficacious interventions to manipulate and control autophagy to improve treatments against cancer. 

## Figures and Tables

**Figure 1 cancers-13-05622-f001:**
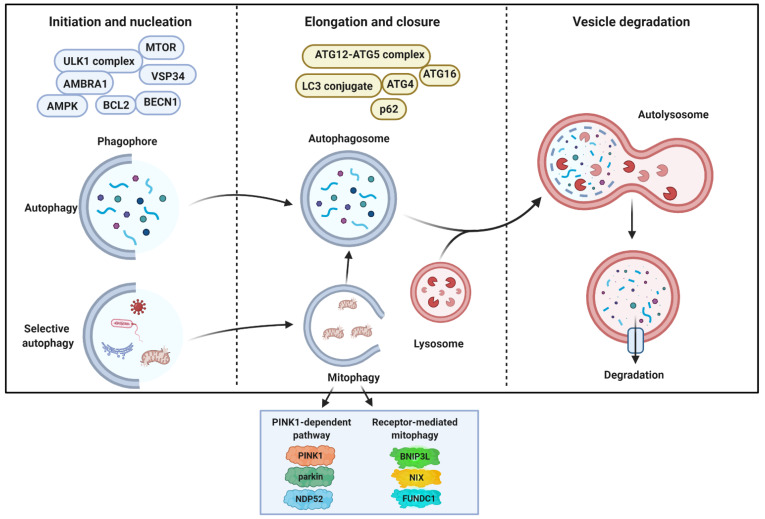
Molecular aspects of autophagy. Autophagy is a catabolic process used by eukaryotic cells to maintain cellular homeostasis and cellular and organellar quality control in response to multiple forms of stress. Different factors are involved during the main stages of autophagy. During the initiation step, the Unc-51-like kinase 1 (ULK1) serine threonine kinase complex receives input from the mechanistic target of rapamycin (MTOR) and from the kinase 5′ AMP-activated protein kinase (AMPK). Meanwhile, MTOR is the main autophagic repressor; AMPK is an autophagy activator. The ULK1 complex then activates nucleation of the phagophore by activating members of class III phosphatidylinositol 3 kinase (PI3KC3) complexes, such as AMBRA1, VSP34, and BECN1. BCL2 regulates the activity of BECN1. Ubiquitin-like conjugation systems ATG12-ATG5-ATG16 and LC3-ATG4 are necessary for the maturation of the vesicle. Other proteins, such as p62, complete the autophagosome formation. The last phase is characterized by fusion of the autophagosome with lysosome and the degradation of the autophagic cargo by acidic hydrolases.

**Figure 2 cancers-13-05622-f002:**
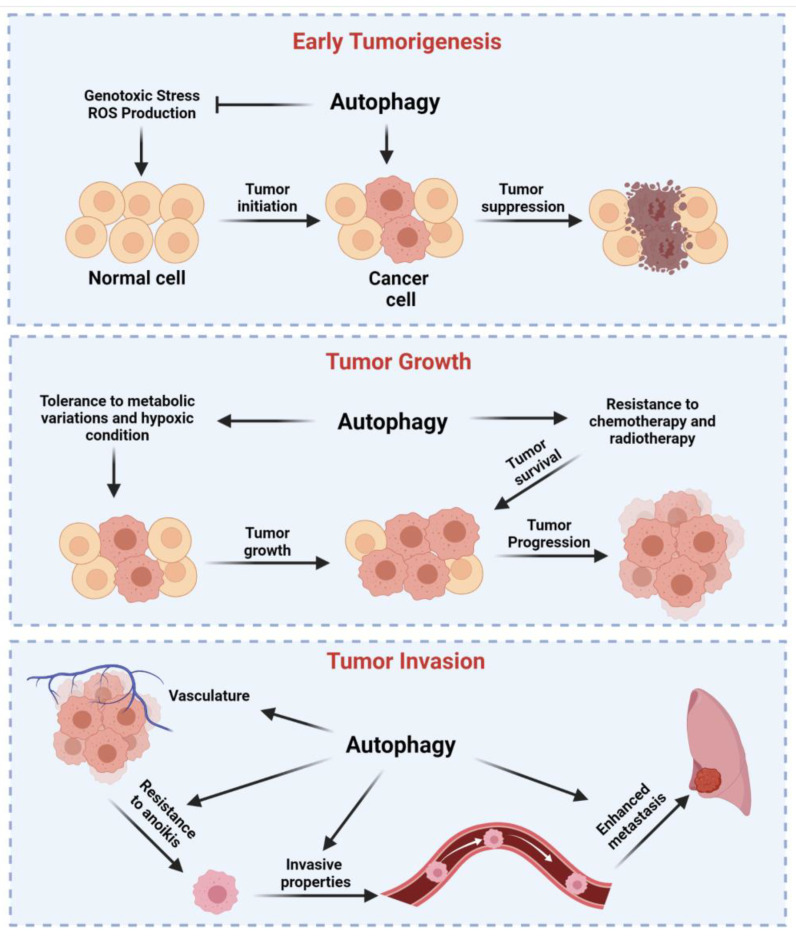
Autophagy role during the different stages of tumorigenesis. Different evidence suggests that, during tumor initiation, autophagy has anti-cancer properties since it protects from agents that may provoke DNA damaging. Furthermore, autophagy has been found to increase the mortality of cancerous cells. Opposingly, autophagy drives the tumor growth of established tumors, enhances cancer cell survival in unfavorable conditions (such as nutrient restriction and hypoxic conditions), and permits cancer cells to evade escape from cancer therapy. Additionally, autophagy promotes the tumor cell metastasis by increasing vascularization and invasive properties and by reducing cell death events against circulating cancer cells.

**Figure 3 cancers-13-05622-f003:**
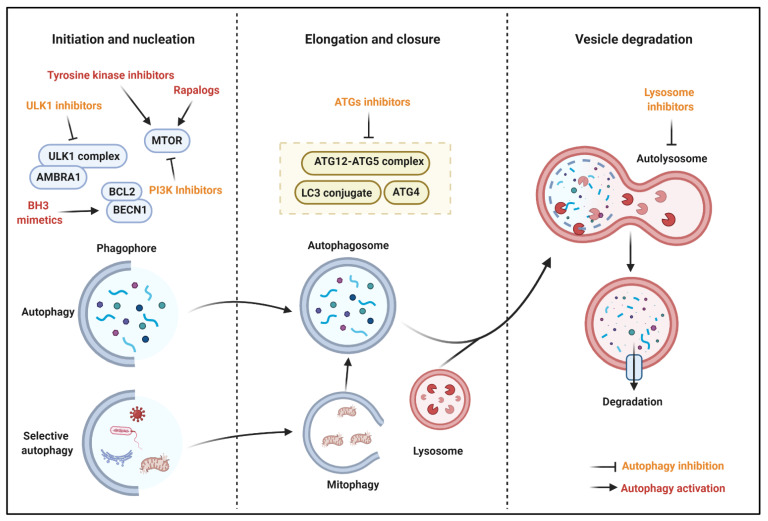
Autophagic modulation for cancer. Autophagy represents a promising opportunity for cancer therapy. Considering the dual role of autophagy during the different phases of cancer, it is not surprising that both activators and inhibitors are feasible anti-cancer compounds. Tyrosine kinase inhibitors (such as sorafenib), rapalogs (everolimus and temsirolimus), and BH3 mimetics (such as obatoclax and sabutoclax) activate autophagy during the early phase. Opposingly, ULK1 (such as SBI-0206965, ULK-1001, and MRT67307) and PI3K inhibitors (such as wortmannin and 3-methyladenine) interfere with this autophagy phase. ATG inhibitors (like NSC185058) block autophagy by preventing the elongation and closure of the autophagic vesicle. Finally, lysosome inhibitors (like chloroquine and derivatives) avoid the fusion of the lysosome with autophagosome or block the activity of lysosomal hydrolases.

**Table 1 cancers-13-05622-t001:** Autophagy-related gene signature in breast, colorectal, bladder, and lung cancer.

Cancer Type	Autophagy-Related Gene Identified	Function	References
Breast	BCL2, BIRC5, EIF4EBP1, ERO1L, FOS, GAPDH, ITPR1, and VEGFA	Upregulated levels have been associated with increased survival	[118]
	EIF4EBP1 and ATG4A	Risk associated genes in advanced breast cancer subgroups (stage III–IV)	[119]
	BAG1, MAP1LC3A, and SERPINA1	Protective genes in advanced breast cancer subgroups (III–IV)	[119]
	VPS35	Oncogenic and prognostic factor	[120]
Colorectal	CX3CL1, ULK3, CDKN2A, NRG1, ATG4B, GAA, RGS19, DDIT3, GRID1, DAPK1, and SERPINA1	Associated with the immune microenvironment of CRC	[121]
	LC3 and BECN1	Low value has been associated with a good response to treatment and a good survival prognosis	[122]
	LC3A	Increased expression was linked to metastasis and a worse prognosis in patients with stage IIA-III colorectal adenocarcinomas	[115]
Lung	ATG10	High expression levels were associated with unfavorable prognosis in non-small cell lung cancer	[123]
	LC3A	Increased levels associated with a poor overall survival	[124]
	ATG16, DRAM, ATG12	High expression in lung adenocarcinoma	[117]
	LAMP2, ATG5, LC3	High expression in squamous cell lung cancer	[117]
Bladder cancer	ATG12, FYCO1, TECPR1, and ULK1	Reduced expression levels in bilharzial bladder cancer	[125]
	ATG4B, DRAM, ATG5, PTEN, and ULK	High expression associated with higher survival	[126]

**Table 2 cancers-13-05622-t002:** Clinical trials reporting results on CQ and HCQ use in cancer. The entire list and full information for clinical trials investigating CQ and HCQ use in cancer are reported at: https://clinicaltrials.gov/ct2/results?cond=cancer&term=chloroquine+&cntry=&state=&city=&dist=, accessed on 30 October 2021 and https://clinicaltrials.gov/ct2/results?cond=cancer+hydroxychloroquine&term=&cntry=&state=&city=&dist=, accessed on 30 October 2021, respectively.

Treatment	Tumor Type	Study Phase	Results	Clinical Trials.gov Identifier	Publication
HCQ 200 mg twice daily with docetaxel 75 mg/m^2^ intravenously (IV) every 21 days on day 1 of the treatment cycle. A cycle is defined as 21 days.	Metastatic prostate cancer	Phase II	Lack of efficacy	NCT00786682	NONE
Erlotinib 150 mg per day with HCQ given at escalating doses of 400, 600, 800, and 1000 mg per day.	Advanced non-small cell lung cancer (NSCLC)	Phase II	Low efficacy	NCT01026844	[287]
Daily administration of HCQ (400 mg) and rapamycin (2 mg) in combination with metronomic chemotherapy.	Refractory metastatic solid tumors	Phase I	Benefit rate of 84% in cohort patients	NCT00909831	NONE
HCQ (200 and 400 mg daily) with rapamycin 2 mg twice a week.	Lymphangioleiomyomatosis	Phase I	Limited response	NCT01687179	[288]
HCQ (1200 mg daily) with rapamycin 2 mg twice a week.	Pancreatic cancer	Phase II	Moderate response	NCT01978184	NONE
HCQ (400 mg) with rapamycin 2 mg daily for 2 treatment cycles composed of 28 days each.	Soft tissue sarcoma	Phase II	Partial response	NCT01842594	[289]
HCQ (200 mg daily) with temozolomide (150–200 mg/m^2^ IV every 28 days) on 1st day of radiotherapy. After 6 weeks, 4 weeks of HCQ alone daily.	Brain and central nervous system tumors	Phase I	Efficacious to block autophagy	NCT00486603	[286]
HCQ (200 mg daily) with temozolomide (75 mg/m^2^ IV every 28 days) on 1st day of radiotherapy. After 6 weeks, 4 weeks of HCQ alone daily.	Brain and central nervous system tumors	Phase II	Limited response	NCT00486603	[286]
Daily HCQ (200, 400, 600, 800, 1000, or 1200 mg) after first dose of gemcitabine (10 mg/m^2^ IV) on days 1 and 15, prior to surgical resection.	Subjects with high risk stage IIb or III adenocarcinoma of the pancreas	Phase I/II	Safe and well tolerated with encouraging oncologic outcomes	NCT01128296	[284]
HCQ 200–600 every day. Bortezomib 1.0–1.3 mg/m^2^ IV at days 1, 4, 8, and 11 of each 21 day cycle.	Multiple myeloma and plasma cell neoplasm	Phase I	Partial response	NCT00568880	[290]
Vorinostat (400 mg daily) with gemcitabine (1000 mg/m^2^) and abraxane (125 mg/m^2^) IV on days 3, 10, 17, 31, 38, and 45 as an intravenous infusion.	Breast cancer	Phase II	Moderate response	NCT00365599	[291]
CQ 150 mg 1 h prior to the radiation treatment (30 Gy in 10 daily fractions from Monday to Friday). CQ treatment continues for 28 days.	Brain metastasis	Phase II	Limited response	NCT01894633	[292]
HCQ low (400 mg twice per day) and high (600 mg twice per day) dose for 1 month prior to surgical removal of the ductal carcinoma in situ lesion.	Metastatic pancreatic cancer	Phase II	Lack of efficacy	NCT01273805	[293]
Oral imatinib mesylate (IM, 400–600 mg) daily and oral HCQ (400 mg) twice daily. Treatment repeats every 4 weeks for up to 12 months.	Chronic myeloid leukemia	Phase II	Well tolerated; clinical advantage for 48 weeks in IM/HCQ group	NCT01227135	[294]

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
