# Peer review of "Understanding the Role of Autophagy in Cancer Formation and Progression Is a Real Opportunity to Treat and Cure Human Cancers"

_cancers, 2021, doi:10.3390/cancers13225622_

Round 1

Reviewer 1 Report

The authors have provided a complete review to understand the role of autophagy in health and disease as well as in cancer. The authors have focused on the dual role of autophagy both as tumor suppressor and as promoter and report the therapeutic opportunities based on the modulation of autophagy process. The manuscript is comprehensive and well written.

Some minor points that may be addressed for more enrichment of the article.

  1. The authors did not mention any details on the role of autophagy in cancer stem cells. Since autophagy regulates the therapy resistance, it will be good to know its regulation of cancer stemness.
  2. Current studies showed the importance of immunotherapy in cancer prevention. Does autophagy play a role in controlling immune responses in cancer and help in sensitization to the immunotherapy?

Author Response

Reviewer 1

Comments and Suggestions for Authors

The authors have provided a complete review to understand the role of autophagy in health and disease as well as in cancer. The authors have focused on the dual role of autophagy both as tumor suppressor and as promoter and report the therapeutic opportunities based on the modulation of autophagy process. The manuscript is comprehensive and well written.

Some minor points that may be addressed for more enrichment of the article.

  1. The authors did not mention any details on the role of autophagy in cancer stem cells. Since autophagy regulates the therapy resistance, it will be good to know its regulation of cancer stemness.

We thank the reviewer for the suggestion. Now we have included a section regarding the importance of cancer stem cells for autophagy and cancer progression.

  1. Current studies showed the importance of immunotherapy in cancer prevention. Does autophagy play a role in controlling immune responses in cancer and help in sensitization to the immunotherapy?

We thank the reviewer for the suggestion. Now, we have included a section entitled “Autophagy and tumor immune response” and added the relevant findings about the importance of immunotherapy for cancer treatment and prevention.

Reviewer 2 Report

Patergnani et al have done a fairly decent job with this review of autophagy in cancer. Particularly well done is the part on inhibitors and the current clinical aspirations in using these drugs. In terms of improvement, some of the mechanistic information in section 4 could be better organized and additionally improved with a few updated references; some of these citations are older and more recent substantial updates with regards to some mechanisms are in the literature but not cited.

Lastly, a simple table that summarizes some of the clinical trials data would be helpful.

Author Response

Reviewer 2

Comments and Suggestions for Authors

Patergnani et al have done a fairly decent job with this review of autophagy in cancer. Particularly well done is the part on inhibitors and the current clinical aspirations in using these drugs. In terms of improvement, some of the mechanistic information in section 4 could be better organized and additionally improved with a few updated references; some of these citations are older and more recent substantial updates with regards to some mechanisms are in the literature but not cited.

We thank the reviewer for the observation. We modified and improved the section 4 as requested. We also added new findings and relative references.

Lastly, a simple table that summarizes some of the clinical trials data would be helpful.

We thank the reviewer for the suggestion. Now, we enclosed a table that summarizes the clinical trials. Since the clinical trials in which autophagy modulators are numerous (22 for chloroquine and 77 for hydroxychloquine), we only included the clinical trials in which the efficacy was reported.

Reviewer 3 Report

In this manuscript entitled ‘Understanding the Role of Autophagy in Cancer Formation and Progression is a Real Opportunity to Treat and Cure Human Cancers’, the authors aim to provide a comprehensive review on the role of autophagy in cancer, covering fundamental scientific findings and clinical implications. A good review article on this topic will appeal to a broad readership. To improve the scientific quality of the manuscript, I have to point out some issues for the authors to address.

Comments:

Major

1. Abstract

Autophagy should have dual roles, prosurvival or prodeath, in cancer, depending on the context. The abstract should cover both.

2. General aspects of autophagy and molecular mechanisms

2-1. Authors should expand the general introduction to macroautophagy in the 2nd para, as macro is the primary form of autophagy.

2-2. References are missing in the 3rd para.

2-3. References are missing in the 2nd last para.

2-4. In the last para, the authors addressed that ‘characterize to sequester and degrade specific intracellular material, such as proteins (proteinphagy), lipids (lipophagy)’. However, lipophagy should refer to autophagic degradation of lipid droplets (subcellular organelles) but not lipids (materials).

2-5. In the last para, the authors addressed that ‘This selective catabolic process was firstly discovered during the differentiative process occur-ring in red blood cell’. Authors should cite the original papers of this finding. In addition, mitophagy should be firstly described by Lewis M and Lewis W in 1915 and firstly observed by the EM in hepatic cells by Ashford T and Porter KR in 1962, although the term ‘mitophagy’ was firstly used in 2006.

2-6. The latest advances in mitophagy research, including the regulation by IMM and matrix proteins, should be covered in the last para.

3. Autophagy in health and disease

3-1 References are missing in many sentences of the 1st para.

3-2. Authors should cite and summarize the findings of the original research rather than rephrasing other review articles.

3-3. Authors should summarize the pathophysiological roles of autophagy in diseases in separate para based on the condition.

4. Dual role of autophagy in Cancer regulation

4-1. In this section, the authors correctly summarized the evidence regarding tumor-suppressive and tumor-promoting roles of autophagy in cancers. In addition to listing these facts, authors should also comment on why autophagy exhibits dual roles in cancer, by discussing those inconsistent findings in the same cancer type. This would raise readers’ thinking about when autophagy activators or inhibitors should be applied during cancer treatment before further reviewing the autophagy-targeted therapies.

4-2. Authors should provide a Table listing the role of autophagy in different cancer types relative to the stage.

5. Cancer Clinical Trials

In this section, authors should provide a Table listing the outcomes of major completed clinical trials and the progress of major ongoing trials.

Minor

  1. There are too many grammar errors contained in the main text. English proofreading is required.
  2. Authors need to replace obscure words, e.g. regulate, influence and affect, by increase, decrease, promote, suppress, etc. Also, to replace ‘a significant percentage of’ by precise numbers.

Author Response

Reviewer 3

Comments and Suggestions for Authors

In this manuscript entitled ‘Understanding the Role of Autophagy in Cancer Formation and Progression is a Real Opportunity to Treat and Cure Human Cancers’, the authors aim to provide a comprehensive review on the role of autophagy in cancer, covering fundamental scientific findings and clinical implications. A good review article on this topic will appeal to a broad readership. To improve the scientific quality of the manuscript, I have to point out some issues for the authors to address.

Comments:

Major

  1. Abstract

Autophagy should have dual roles, prosurvival or prodeath, in cancer, depending on the context. The abstract should cover both.

We agree with the reviewer. We modified the abstract as requested.

  1. General aspects of autophagy and molecular mechanisms

2-1. Authors should expand the general introduction to macroautophagy in the 2nd para, as macro is the primary form of autophagy.

We expanded the part about macroautophagy as requested.

2-2. References are missing in the 3rd para.

We thank the reviewer for the observation. We included new references.

2-3. References are missing in the 2nd last para.

We thank the reviewer for the observation. We included new references.

2-4. In the last para, the authors addressed that ‘characterize to sequester and degrade specific intracellular material, such as proteins (proteinphagy), lipids (lipophagy)’. However, lipophagy should refer to autophagic degradation of lipid droplets (subcellular organelles) but not lipids (materials).

We thank the reviewer for the observation. We corrected it.

2-5. In the last para, the authors addressed that ‘This selective catabolic process was firstly discovered during the differentiative process occur-ring in red blood cell’. Authors should cite the original papers of this finding. In addition, mitophagy should be firstly described by Lewis M and Lewis W in 1915 and firstly observed by the EM in hepatic cells by Ashford T and Porter KR in 1962, although the term ‘mitophagy’ was firstly used in 2006.

We thank the reviewer for the suggestion. We modified the part about mitophagy as requested.

2-6. The latest advances in mitophagy research, including the regulation by IMM and matrix proteins, should be covered in the last para.

We thank the reviewer for the suggestion. We improved the part about mitophagy.

  1. Autophagy in health and disease

3-1 References are missing in many sentences of the 1st para.

We thank the reviewer for the observation. We included new references

3-2. Authors should cite and summarize the findings of the original research rather than rephrasing other review articles.

We thank the reviewer for the suggestion. References of original articles have been now included in the manuscript.

3-3. Authors should summarize the pathophysiological roles of autophagy in diseases in separate para based on the condition.

As requested, we separated the role of autophagy in the different diseases in separated paragraphs.

  1. Dual role of autophagy in Cancer regulation

4-1. In this section, the authors correctly summarized the evidence regarding tumor-suppressive and tumor-promoting roles of autophagy in cancers. In addition to listing these facts, authors should also comment on why autophagy exhibits dual roles in cancer, by discussing those inconsistent findings in the same cancer type. This would raise readers’ thinking about when autophagy activators or inhibitors should be applied during cancer treatment before further reviewing the autophagy-targeted therapies.

We improved the section as requested.

4-2. Authors should provide a Table listing the role of autophagy in different cancer types relative to the stage.

We thank the reviewer for the suggestion. We believe that a table summarizing the role of autophagy in a specific tumor relative to the stage would be very useful. Unfortunately, at today it does not exist specific investigations in which the autophagy role has been well associated to the exact stage of a specific cancer type. Investigations are limited to verify the sensitiveness of cancer subtypes to a curative treatment composed of chemotherapy agent and autophagy modulators (PMID: 31488317 for breast cancer; PMID: 31383875 for colorectal cancer). Not less important, all these studies have been performed in cancer cell lines and not in primary cells obtained by patients affected from different cancer subtypes. Opposite, several investigations analyzed the autophagy-related gene signature in samples obtained from individual affected by various cancer types. For example, gain in expression in 8 autophagy-related gene has been associated with overall survival in breast cancer (PMID: 25620657). Similarly, the expression levels of 7 autophagy-related genes have been found upregulate in bladder cancer patients (PMID: 28381171). In addition, a gene signature of 11 autophagy genes were identified in colorectal cancer (PMID: 34699544) (Table 1). Once more, an autophagy-associated gene signature has been also found significantly related to survival in lung cancer (PMID: 31811814). Interestingly, in certain cases, investigators also analyzed the autophagy levels in diverse cancer subtype (PMID: 26600927 31811814 20876316) (Table 1). However, the primary aim of these studies is to identify new factors that could help prognosis and therapy, thereby lacking to associate the level of expression of the autophagy gene investigated to the effective autophagy function and levels at the time of sample collection.
For all these reasons, we cannot include a table summarize the autophagy involvement in different cancer subtypes. Nevertheless, we include a table summarizing the different gene signatures in 3 different cancer types, describing the role of these signatures and the possible association among the diverse cancer subtypes.

  1. Cancer Clinical Trials

In this section, authors should provide a Table listing the outcomes of major completed clinical trials and the progress of major ongoing trials.

We thank the reviewer for the suggestion. Now, we have enclosed a table that summarizes the clinical trials. Since the clinical trials in which autophagy modulator are numerous (22 for chloroquine and 77 for hydroxychloquine), we only included the clinical trials in which the efficacy was reported.

Minor

  1. There are too many grammar errors contained in the main text. English proofreading is required.
  2. Authors need to replace obscure words, e.g. regulate, influence and affect, by increase, decrease, promote, suppress, etc. Also, to replace ‘a significant percentage of’ by precise numbers.

We performed an accurate English proofreading of the manuscript and replaced obscure words.

Round 2

Reviewer 3 Report

The manuscript has been significantly improved by taking review comments. It is currently qualified for publication.